# Novel Characterizations of Effective SIFs and Fatigue Crack Propagation Rate of Welded Rail Steel Using DIC

**Xiu-Yang Fang [1], Jian-En Gong [1], Wei Huang [1], Jia-Hong Wu [1] and Jun-Jun Ding [1,2,\*]**

[1] School of Mechanical Engineering, Southwest Jiaotong University, Chengdu 610031, China
[2] MOE Key Laboratory of High-speed Railway Engineering, Southwest Jiaotong University, Chengdu 610031, China
\* Correspondence: dingjunjun@swjtu.cn

**Abstract:** The fatigue crack growth-rate test of rail head, waist, and bottom material for U71Mn welded rail was carried out. Digital image correlation (DIC) was used to capture the full-field displacement. The crack-tip position was accurately obtained based on the full-field displacement data, and an accurate crack-tip opening displacement (CTOD) measurement point was found. The CTOD values of the welded rail head under overloaded and unloaded condition were extracted, and the area size of elastic CTOD and plastic CTOD was obtained. According to COD data under different experimental conditions, the corresponding crack opening force was extracted, the crack opening function introduced based on the Elber model, and a calculation method of effective stress-intensity factors (SIFs) considering the plasticity-induced crack closure proposed. The results in this paper provide some references for accurately assessing the fatigue life of welded rail.

**Keywords:** U71Mn rail; fatigue crack propagation; effective SIFs; crack closure effect; DIC





## 1. Introduction

As an important bearing structure for railway lines, railway rails usually have to withstand complex rolling, sliding, and impact loads during service, and the working conditions are very harsh. As the health and integrity of rail lines are related to the operational safety of rail vehicles, it is crucial to accurately and quantitatively evaluate the service performance of rails under the action of harsh cyclic external loads, that is, fatigue lifetime. At present, welded gapless rails are generally used in high-speed railway lines, and welded joints are usually the mechanically weak areas of the rail lines; therefore, accurate quantitative evaluation of their anti-fatigue crack initiation and propagation ability has important theoretical and engineering significance.

Currently, for metallic materials, the stress-intensity factor (SIF), $K$ parameter, is often used to quantitatively characterize the fatigue crack propagation susceptibility of a mechanical structure.

In 1963, Paris et al. [1] proposed a mathematical model for quantitatively characterizing the resistance of materials to fatigue crack propagation, which is still currently widely used; its expression is shown in Equation (1):

$$\frac{da}{dN} = C(\Delta K)^m$$
$$\Delta K = K_{\max} - K_{\min} \tag{1}$$

where $C$ and $m$ are material-dependent constants.

Subsequently, Elber [2] found that under the condition of tensile stress, crack surfaces usually contact each other, which results in fatigue crack closure, which has a certain influence on the propagation of the fatigue crack. Additionally, the effective stress-intensity

factor $\Delta K_{eff}$ was proposed, and then the classical Paris model was modified and improved. The improved model is given as Equation (2):

$$\frac{da}{dN} = C(\Delta K_{eff})^m = C(U\Delta K)^m$$
$$U = \frac{1-f}{1-R}f = \frac{K_{op}}{K_{max}} = \frac{F_{op}}{F_{max}} \tag{2}$$

where $U$ is the crack opening function and $F_{op}$ is the crack opening force.

The research of Elber [2] shows that the plastic-induced crack closure of the crack tip is often produced in the fatigue crack propagation, which makes the calculated parameter $\Delta K$ according to the traditional method deviate from the actual $\Delta K$. The classical Paris model does not consider the effect of the crack-tip plasticity-induced crack closure effect, and it is still an international academic issue to accurately characterize the effective stress-intensity factor $\Delta K_{eff}$ under consideration for the crack-tip plasticity-induced crack closure effect. At present, research on the effect of plasticity-induced crack closure locally and abroad mainly includes both numerical simulation [3–5] and experimental methods [6–8]. Among the experimental methods, DIC technology is a noncontact, high-precision method for full-field displacement, deformation, and stress measurement, which is increasingly used in various fields because of its extremely relaxed test environment requirements and its advantages of full-field measurement, strong anti-interference ability, and high measurement accuracy. The DIC method has gained many impressive and successful applications in the measurement of surface deformation in various materials and structures, in the characterization of mechanical and physical parameters, and in the verification of mechanical theory and finite element analysis.

The first difficulty in characterizing the crack-tip displacement field by DIC is how to accurately determine the real-time crack-tip position during fatigue crack propagation. Regarding the method for determining the precise location of the crack tip, domestic and foreign researchers have carried out several investigations. Mokhtarishirazabad et al. [9] determined crack-tip location by increasing the DIC image resolution followed by direct observation at a higher magnification. Avtaev et al. [10] used the distribution pattern of $e_{yy}$ (vertical strain) on the crack extension to identify the tip location. Vasco-Olmo et al. [11–13] used the distribution law of vertical displacement with the y-coordinate to determine the y-coordinate of the crack tip and recorded its vertical displacement value and then determined the x-coordinate of the crack tip based on the distribution law of vertical displacement with the x-coordinate and the recorded vertical displacement value to determine the crack-tip coordinates. Gao et al. [14] located subimages containing cracks from the crack-tip image based on the image grayscale-difference (GSD) distribution, then extracted the crack contour from the subimages using the maximum entropy threshold segmentation method, and lastly extracting the crack skeleton image from the crack contour image by the morphological refinement algorithm to determine the location of the crack tip. Among many methods, the DIC method can used to directly obtain accurate displacement field data for the specimen under external load.

The crack-tip opening displacement (CTOD) is the difference in vertical displacement at a certain distance from the crack tip, and the magnitude of the CTOD value reflects the cracking resistance of the material. Khor et al. [15] measured CTOD using the δ5 method [16] and the DIC method, where δ5 is the CTOD measurement at two points on the surface of the specimen initially 5 mm apart. Their work used bent specimens of austenitic stainless steel notched on one side and compared the results with the obtained CTOD measurements and measured the fracture toughness using the silicon replication and clamping methods. The measured CTOD values were not consistent with the crack-tip opening values originally defined by Wells [17]. Samadian et al. [18] proposed a new method to measure the CTOD of the entire crack glyph by measuring the 3D profile of the notched surface using 3D DIC, which was validated by silicone replica measurements and finite element analysis. Ktari et al. [19] studied the fatigue crack expansion of AISI 4130 forging steel at load ratios of 0.1 and 0.7. They proposed a two-dimensional DIC method based on

ΔCTOD to characterize fatigue crack expansion and concluded that ΔCTOD can be used as a feasible parameter to characterize fatigue crack expansion.

ΔCTOD can be further subdivided into $\Delta CTOD_{pl}$ (plastic CTOD) and $\Delta CTOD_{el}$ (elastic CTOD). Several results have been presented to show that plastic CTOD parameters can be used to qualitatively characterize the driving force of fatigue crack extension resistance in materials [20]. Marques et al. [20] calculated $\Delta CTOD_{pl}$ and $\Delta CTOD_{el}$ for several instances using the finite element method and established a simple criterion that defines the boundary of the small-scale yielding (SSY) region to avoid the ineffective use of the LEFM parameter ΔK as a characterization parameter for the fatigue crack expansion rate. Andre Prates et al. [21] conducted a numerical study of the effect of material parameters and load variability on plastic CTOD results and established a relationship between the most influential parameters and the range of plastic CTOD.

The measurement of crack opening force is an important aspect of the study of crack closure effect, and since the crack closure effect has been proposed, scholars proposed several methods to measure $K_{op}$ and $P_{op}$, which can be divided into direct and indirect methods [22]. The direct methods include the observation method [23], photography method [24], surface strain method [25], etc.; the indirect method mainly obtains the crack tension force by studying the flexibility curve of the specimen. As early as 1970, Elber [26] used experimentally measured flexibility curves to indirectly determine the crack opening force Fop. With the development of measurement technology, the main indirect experimental methods based on the flexibility curve are: crack mouth opening displacement (CMOD) [27], near-crack-tip gage (NCTG), push-rod gage [28–30], and so on. In addition, other techniques such as the acoustic emission (AE), eddy current, potential drop, ultrasonic, and dynamic softness methods have been used [23].

In this paper, CT specimens were prepared from the base material of welded rails and the respective rail head, rail waist, and rail bottom parts of the welded joint. Fatigue crack expansion experiments were conducted by combining related digital image technology. Additionally, the crack-tip position was determined according to the DIC displacement field data law; the CTOD measurement distance between the weld and base material under specific crack length was determined according to the CTOD variation law; the CTOD values under loading and unloading conditions were extracted; the plastic CTOD and elastic CTOD values of the weld specimens were calculated; and finally, the crack opening force of the specimens under each experimental condition was obtained by combining the analysis of the COD data of the crack leading edge, and then a method to calculate the effective IFS was proposed based on Elber model.

## 2. Materials and Experimental Procedure

The as-received material was U71Mn welded rail. To accurately evaluate the fatigue crack propagation resistance of welded rail, this study carried out the fatigue crack propagation-rate experimental test of welded rail under different conditions on the GPS 100 fatigue machine; the experimental CT specimens were obtained as shown in Figure 1a. During the constant amplitude experiment, stress ratio R = 0.1, $F_{max}$ = 1200 N, 20 Hz frequency, and sine wave loading were used; the specimen size is shown in Figure 1b. During the overload experiment, stress ratio R = 0.3, $F_{max}$ = 2000 N, about 100 Hz frequency, and sine wave loading were also used; the specimen size is shown in Figure 1c. When the crack length extended to 14.84 mm, 100% overload was performed (F = 4000 N). The specimen numbers and experimental conditions are listed in Table 1. To insure the reproducibility of the results, pre-experiment was performed before the tests. The mechanical properties of the base material and joint are listed in Table 2 [31,32].

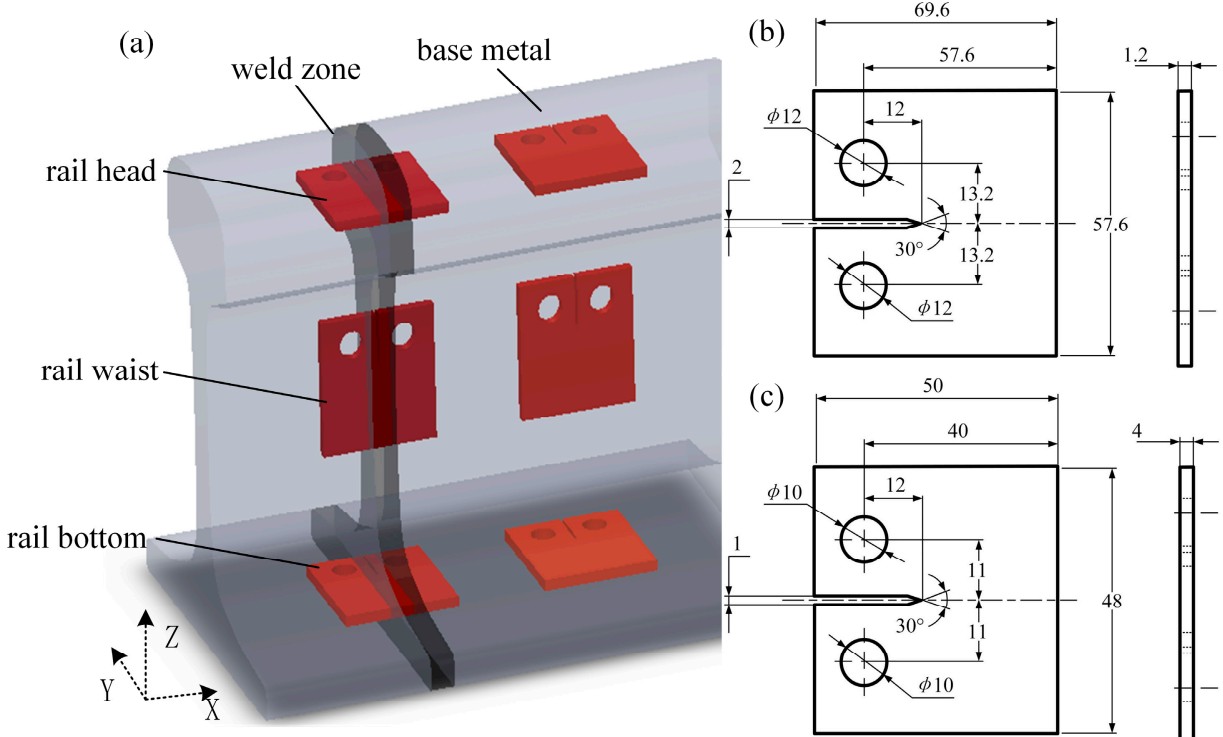

**Figure 1.** (**a**) Schematic diagram of specimen sampling; (**b**) dimensional diagram of unloaded specimen; (**c**) dimensional diagram of overloaded specimen.

**Table 1.** Specimen number and experimental conditions.

| Specimen Number | Specimen Position | Experimental Conditions |
|:---:|:---:|:---:|
| B1 | Rail head of base metal | |
| B2 | Rail waist of base metal | |
| B3 | Rail bottom of base metal | |
| W1 | Welded rail head | Constant amplitude tests |
| W2 | Welded rail waist | |
| W3 | Welded rail bottom | |
| W4 | Welded rail head | Overload test |

**Table 2.** Mechanical properties of the materials [31,32].

| | Base Metals | Welded Joints |
|:---:|:---:|:---:|
| Elastic modulus (GPa) | 210 | 206 |
| Poisson's ratio | 0.3 | 0.3 |
| Yield strength (MPa) | 552 | 608 |

To obtain the full-field information in the vicinity of the crack surface, the displacement field in the vicinity of the crack surface during the fatigue crack propagation was monitored in real time with the help of DIC technology. The test specimen was sprayed with scattered spots on one side for DIC monitoring, and the other side was polished smoothly for real-time crack length observation. The resolution of the DIC test camera is 18.5 μm/pixel. Typical test specimen scattered spots and displacement field results of crack surface are shown in Figure 2.

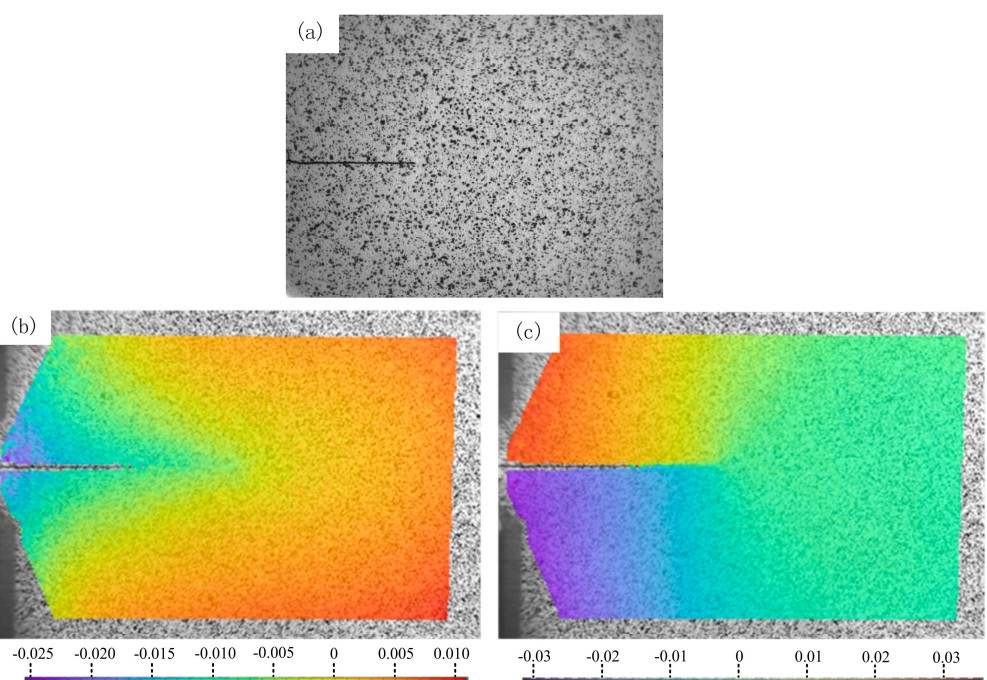

**Figure 2.** Typical specimen scatter DIC treatment: (**a**) scatter; (**b**) transverse displacement contour (mm); (**c**) longitudinal displacement contour (mm).

In the constant amplitude test, non-standard CT specimens were used to facilitate the digital image test, and in view of the sensitivity of the conventional stress-intensity factor to the specimen geometry, the amplitude of the stress-intensity factor for non-standard CT specimens was modified in this study based on the crack stress analysis manual compiled by Tada et al., [33]. The modified equation is shown in Equation (3).

$$\Delta K = \frac{F_{\max} - F_{\min}}{B\sqrt{W}} \times \frac{2+a}{(1-a)^{3/2}} \times 2F_2 \tag{3}$$

where $F_{max}$ and $F_{min}$ are the maximum and minimum loads, respectively, $B$ is the specimen thickness, $W$ is the CT specimen width, $a$ is the crack length, and $F_2$ is the correction function, for the specimens described in this paper.

$$F_2 = 0.618 + 1.89(a/b) - 6.94(a/b)^2 + 8.27(a/b)^3 - 3.25(a/b)^4 \tag{4}$$

where $b$ is the transverse distance from the center of the loading hole to the end face of the specimen ($b$ = 57.6 for the specimen used in this paper).

## 3. Results and Discussion

### 3.1. Determination of Crack-Tip Position

#### 3.1.1. Determination of the y-Coordinate of the Crack Tip

The vertical displacements of specimens B1 and W1 captured by DIC are shown in Figure 3. Since the load on the specimen is symmetric along the vertical direction, the crack tip must be located in the area where the vertical displacement is close to 0, i.e., the red boxed area in Figure 3. From the results, it can be seen that the tip coordinate y = 599 pixel for specimen B1 exhibits crack length a = 25.02 mm, and y = 596 pixel for specimen W1 with crack length a = 24.48 mm.

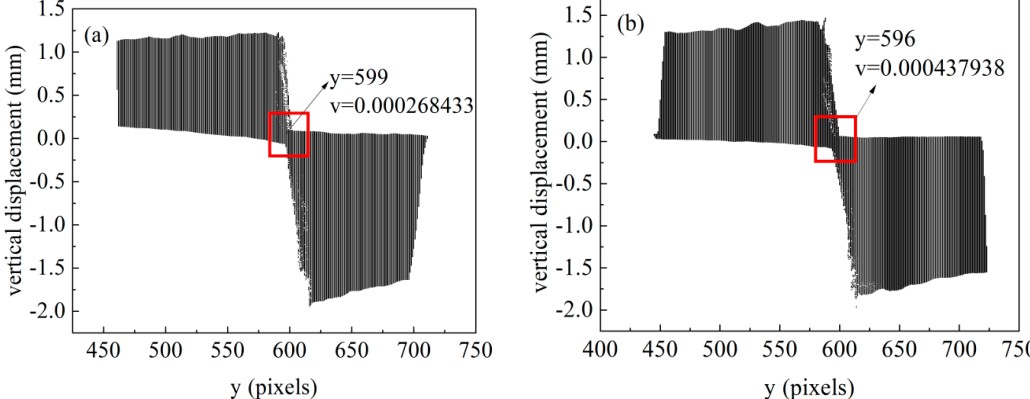

**Figure 3.** Variation in vertical displacement with y-coordinate for specimens B1 and W1 crack areas: (**a**) specimen B1 (*a* = 25.02 mm); (**b**) specimen W1 (*a* = 24.48 mm).

3.1.2. Determination of the x-Coordinate of the Crack Tip

**Method 1**: Minimum vertical displacement to determine the tip x-coordinate.

When the y-coordinate of the crack tip is known, theoretically it is enough to extract the vertical displacement of all the points in this y-coordinate and find the pixel point where the vertical displacement is first close to zero, and then that point is the crack tip. However, owing to the large fluctuation in the data, only the pixel points with positive vertical displacement are considered in this paper. Among these points, the point with the smallest vertical displacement is taken and its x-coordinate is recorded as the crack-tip x-coordinate. The variation in the crack-tip vertical displacement with the x-coordinate for specimens B1 and W1 is at *a* = 25.02 mm and *a* = 24.48 mm, respectively, as shown in Figure 4. Based on this method, the minimum values of the vertical displacement of specimens B1 and W1 are 0.000268433 mm and 0.000437938 mm, corresponding to x-coordinates of *x* = 747 pixels and *x* = 924 pixels, respectively. Combining Figures 3 and 4, it can be seen that the coordinates of the crack tip found by this method are B1 (747, 599), W1 (924, 596).

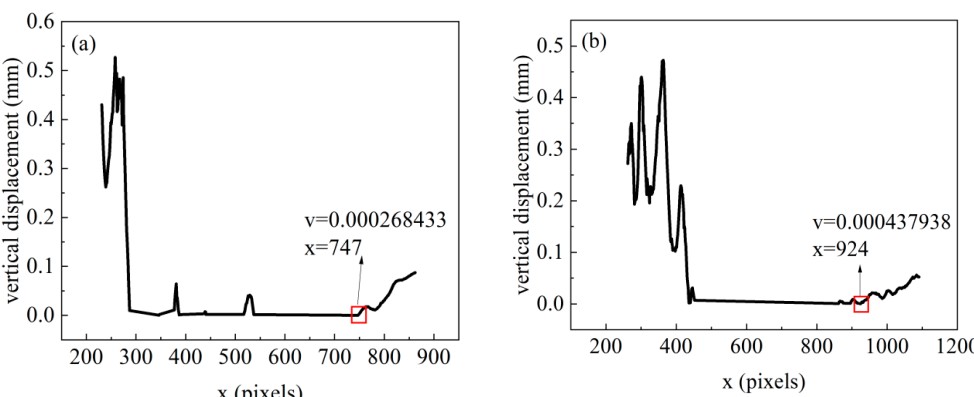

**Figure 4.** Variation in vertical displacement with x-coordinate on the crack line of specimens B1 and W1: (**a**) specimen B1 (*a* = 25.02 mm); (**b**) specimen W1 (*a* = 24.48 mm).

Usually, for metallic materials, a plastic deformation zone is generated at the crack tip, and the generation of the plastic zone will have a certain effect on the continued propagation of fatigue cracks. In this paper, the plastic zone of the crack tip was extracted using the method described in the literature [34] (the flow chart is shown in Figure 5), where the strain field, stress field, and equivalent stress field results were obtained from the displacement field results obtained from the experimental tests, and were calculated as shown in Equations (5)–(7), respectively.

$$\begin{pmatrix} \varepsilon_{xx} \\ \varepsilon_{yy} \\ \varepsilon_{xy} \end{pmatrix} = \begin{pmatrix} \frac{\partial u}{\partial x} \\ \frac{\partial u}{\partial y} \\ \frac{\partial u}{\partial y} + \frac{\partial v}{\partial x} \end{pmatrix} + \frac{1}{2} \begin{pmatrix} \frac{\partial u}{\partial x} & 0 & \frac{\partial v}{\partial y} & 0 \\ 0 & \frac{\partial u}{\partial y} & 0 & \frac{\partial v}{\partial y} \\ \frac{\partial u}{\partial y} & \frac{\partial u}{\partial x} & \frac{\partial v}{\partial y} & \frac{\partial v}{\partial x} \end{pmatrix} \begin{pmatrix} \frac{\partial u}{\partial x} \\ \frac{\partial u}{\partial y} \\ \frac{\partial v}{\partial x} \\ \frac{\partial v}{\partial y} \end{pmatrix} \tag{5}$$

$$\begin{aligned} \sigma_{xx} &= \frac{E}{1-v^2}(\varepsilon_{xx} + v\varepsilon_{yy}) \\ \sigma_{yy} &= \frac{E}{1-v^2}(\varepsilon_{yy} + v\varepsilon_{xx}) \\ \sigma_{xy} &= \frac{E}{1+v}\varepsilon_{xy} \end{aligned} \tag{6}$$

$$\sigma_{eq} = \frac{1}{\sqrt{2}}\sqrt{(\sigma_{xx} - \sigma_{yy})^2 + \sigma_{xx}^2 + \sigma_{yy}^2 + 6\sigma_{xy}^2} \tag{7}$$

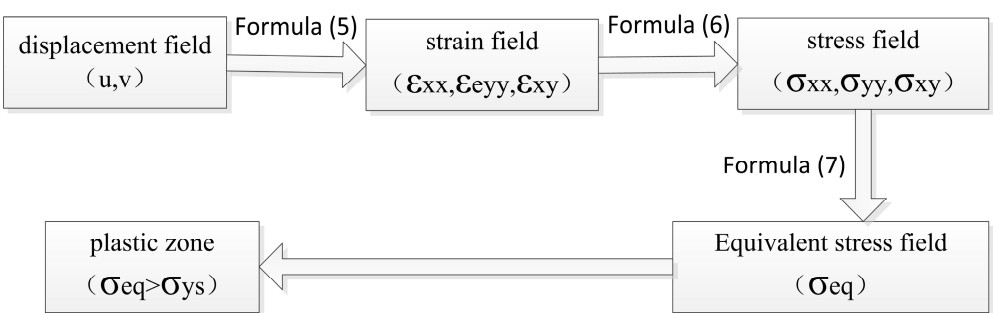

**Figure 5.** Flow chart showing how DIC was used to solve the stress field by the displacement field of the crack surface.

Figure 6 shows the extracted plastic zone contours of the base rail head and the welded rail head. As can be seen from Figure 6, for specimen B1 $x$ = 747 pixels and for specimen W1 $x$ = 924 pixels, located behind the plastic zone; there is a large deviation in finding the crack-tip $x$-coordinate using this method.

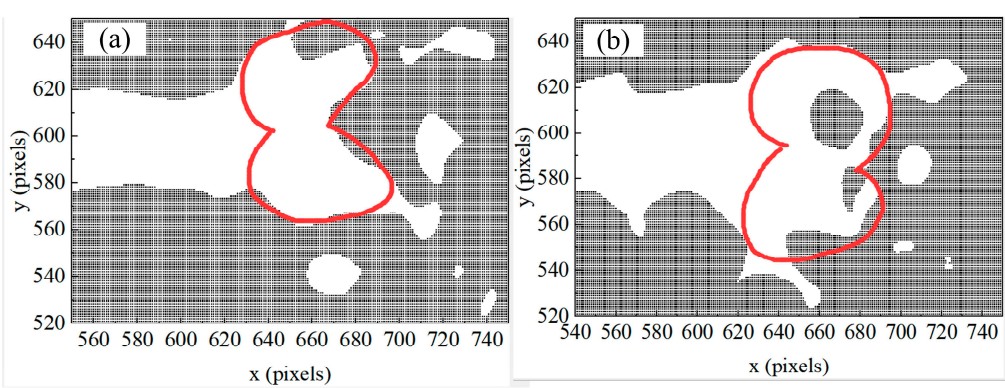

**Figure 6.** Contours of the plastic zone at the crack tip of specimen B1 and W1: (**a**) B1-a = 25.02 mm; (**b**) W1-a = 24.48 mm.

**Method 2**: Plastic zone profile retraction $r_p$ method to determine the tip x-coordinate.

For the plane stress state, Irwin [35] proposed a formula concerning the plastic zone $r_p$ at the crack tip as given in Equation (8) ($r_p$ represents the length of the plastic zone on the crack line, as shown in Figure 7):

$$r_p = \frac{1}{2\pi}\left(\frac{K_{max}}{\sigma_{p0.2}}\right)^2 \tag{8}$$

where $\sigma_{p0.2}$ is the yield strength of the material.

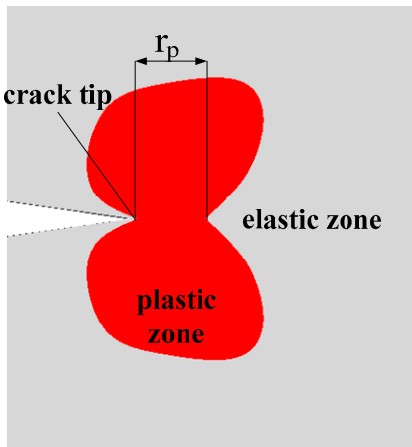

**Figure 7.** Schematic diagram of plastic zone $r_p$.

The $r_p$ values of 0.676421587 mm (37 pixels) and 0.647278593 mm (35 pixels) for specimen B1 a = 25.02 mm and specimen W1 a = 24.48 mm, respectively, were calculated using Equations (3) and (4). The y-coordinates of the crack tip for specimens B1 and W1 with $a$ = 25.02 mm and $a$ = 24.48 mm were determined in Section 3.1.1 as $y$ = 599 pixels and $y$ = 596 pixels, respectively; then, for specimens B1 and W1, the x-coordinate of the crack tip must be located on the line at $y$ = 599 pixels and $y$ = 596 pixels, i.e., on the red line in Figure 8. If the red line intersects the plastic zone at point A, then point B on the line at a distance of $r_p$ from point A is the location of the crack tip. The relationship $x_B = x_A - r_p$ was used to calculate the x-coordinate of point B. Point B is the exact location of the crack tip.

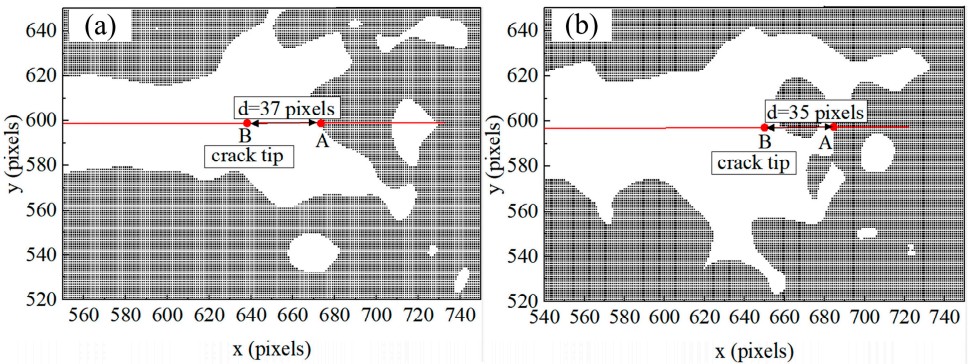

**Figure 8.** Finding the crack-tip x-coordinate according to the plastic zone: (**a**) specimen B1 $a$ = 25.02 mm, (**b**) specimen W1 $a$ = 24.48 mm.

The coordinates of the crack tip at $a$ = 25.02 mm and $a$ = 24.48 mm were found for specimen B1 and for specimen W1 at (636, 599) and (650, 596), respectively, according to the above method.

### 3.2. Determination of CTOD

COD is the crack opening distance $d_y$ in the vertical direction corresponding to the horizontal distance retreat $d_x$ of the crack tip on the crack growth path, defined as shown in Figure 9. After accurately determining the crack-tip positions at different crack growth lengths, there is still some academic controversy on how to accurately obtain the magnitude of the crack-tip opening displacement CTOD values, while the real-time CTOD parameters during fatigue crack extension can be used to characterize the fatigue crack propagation driving force. This paper attempts to obtain accurate CTOD parametric values based on the full-field displacement results obtained from the DIC test.

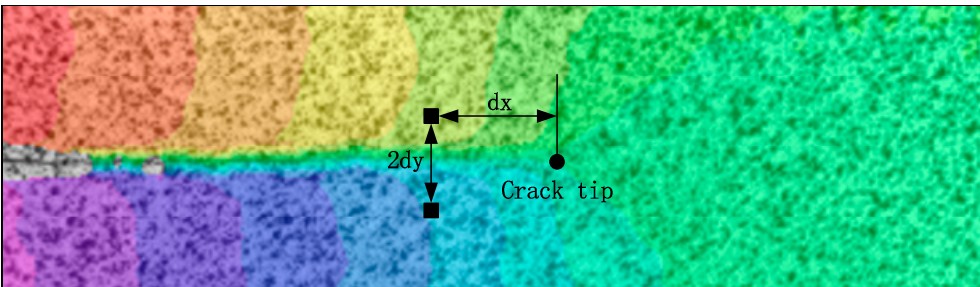

**Figure 9.** COD definition.

In this paper, the COD at different $d_x$ and different $d_y$ were extracted based on the data of the full-field displacement results for specimen B1 and specimen W1 tests, respectively; the results are shown in Figure 10. From Figure 10a,c, it can be found that the larger the $d_x$ is, the larger the COD value is for the same $d_y$, because the greater the distance from the crack tip, the larger the opening magnitude. It can be seen that the COD curves at different $d_y$ are relatively dispersed near the crack tip, and gradually converge to a point as $d_x$ increases. Using this point as a characteristic point, the $d_x$ corresponding to this position is used as the $d_x$ for the crack-tip opening displacement measurement point. In the same way, B1 pattern $d_x$ = 67 pixels (1.24 mm) and W1 pattern $d_x$ = 55 pixels (1.02 mm) are the crack-tip opening displacement transverse-position points. From the results, it can be seen that the $d_x$ of the welded specimen is less than that of the base material, this is because the variation in COD with $d_x$ is affected to some extent by the plastic zone of the crack tip, and the $r_p$ value of the plastic zone shows that the plastic zone of the joint is slightly smaller than that of the base material, so the effect on COD is smaller, which leads to a smaller $d_x$.

From Figure 10b,d, it can be found that with the same $d_x$, the COD first increases gradually with increasing $d_y$ and then stabilizes close to a horizontal line, and for different $d_x$, the COD plateau area appears at the same $d_y$. Defining the position $d_y$ as the $d_y$ of the crack-tip opening displacement and following the same method, the value of the crack-tip opening displacement for specimen B1 is $d_y$ = 29 pixels (0.54 mm) and the value of the crack-tip opening displacement for specimen W1 is $d_y$ = 31 pixels (0.57 mm).

It was found that when the load increased to the maximum, the cracks were fully opened. When the load decreased to a certain value (nonzero), the cracks began to close. This is due to the plastic deformation that occurs at the crack tip; this deformation is irreversible. To study the variation in crack-tip opening displacement during loading and unloading, this paper investigated the weld-overload specimen data. The main study is the change in crack-tip opening displacement when the load is increased from 0.6 to 4 KN and then decreased to 0.6 KN at a 0.2 KN increment. Firstly, find the exact crack-tip location, then extract the COD data to obtain the law, to find the CTOD measurement point location after extracting the CTOD values under different loads, as shown in Figure 11.

As can be seen from Figure 11, the CTOD value is increasing during loading, and the distribution of data points is slightly concave, while the CTOD value is decreasing during load reduction, and the distribution of data points is nearly a straight line, and when the load is reduced to the same level of load at the time of loading, the CTOD is generally greater than the value at the time of loading. Observing the CTOD data of the loaded section, it can be found that the points between point $a$ and point $b$ have a good linear relationship, indicating that the section is in the elastic deformation stage. The data after point $b$ are due to the irreversible plastic deformation at the crack tip of the specimen, so it no longer maintains a linear relationship with the data in section ab. Therefore, a straight line $ab$ can be fitted to the data points between ab and extended to intersect dc at point e. Then, the distance between $ce$ can be taken as plastic CTOD (CTOD$_{pl}$), and the distance between $de$ is elastic CTOD (CTOD$_{el}$). The measurement shows that CTOD$_{pl}$ = 0.1247 mm and CTOD$_{el}$ = 0.6007 mm. The elastic CTOD value is much greater than the plastic CTOD

value. This indicates that the yield strength of the joint is higher and the plastic zone is smaller, which is consistent with reality.

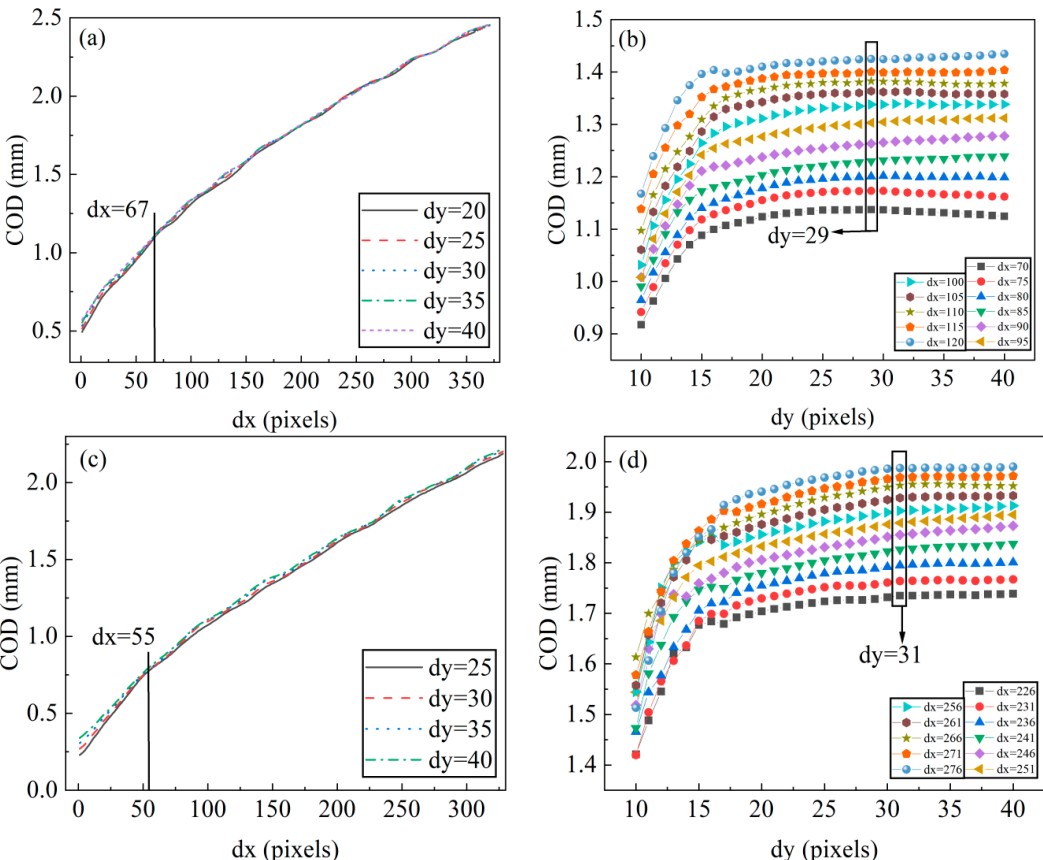

**Figure 10.** Determination of crack-tip opening displacement $d_x$ and $d_y$: (**a**) determination for specimen B1, $a$ = 25.02 mm $d_x$; (**b**) determination for specimen B1, $a$ = 25.02 mm $d_y$; (**c**) determination for specimen, W1 $a$ = 24.48 mm $d_x$, (**d**) determination for specimen W1 $a$ = 24.48 mm $d_y$.

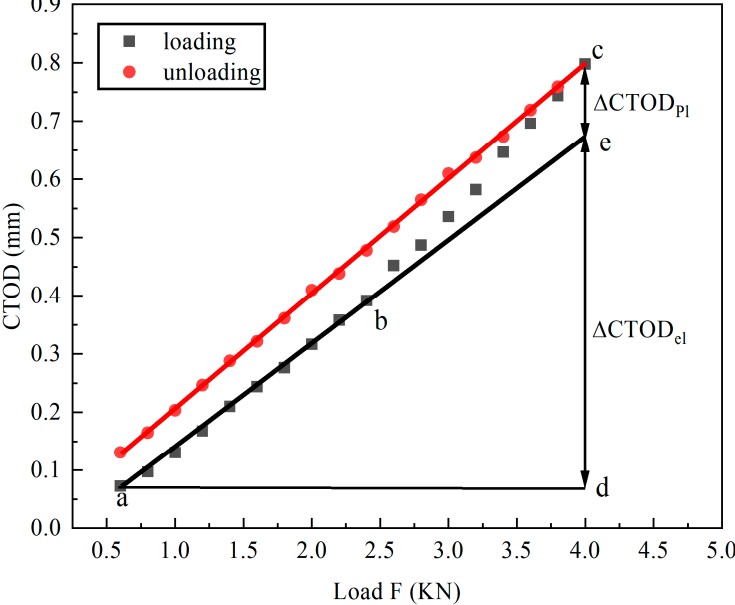

**Figure 11.** Overload weld specimen $a$ = 14.84 mm elastic CTOD and plastic CTOD.

Traditionally, the amplitude of the stress-intensity factor is used as the fatigue crack propagation driving force parameter to characterize the fatigue crack growth rate, which is simple but cannot take into account the effect of crack closure. Plastic CTOD is an important parameter at the crack tip, and there is a linear relationship between the crack expansion rate (da/dN) and $\Delta CTOD_{pl}$ [12]. To a certain extent, plastic CTOD can also characterize the fatigue crack expansion drive, while plastic CTOD takes into account the influence of plasticity-induced crack closure effect at the crack tip, which can more accurately and effectively characterize the fatigue crack expansion drive. In this paper, the work on the extraction of the plastic zone of the crack tip is informative for accurate characterization of the crack extension rate.

*3.3. Characterization of Effective SIFs*

Crack closure exists at the crack tip, and it is important to accurately assess the effect of crack closure on crack expansion to obtain an accurate magnitude for the effective stress-intensity factor. In this paper, the COD data of base material and weld rail head, rail waist, and rail bottom are extracted and the corresponding load $F_{open}$ when the crack opening at different crack lengths is extracted, and then the effective stress-intensity factor is obtained based on Elber theory.

The COD results obtained for different specimens with different crack lengths under different loads are shown in Figure 12. As can be seen from Figure 12, for the parent material, the rail head, rail waist, and rail bottom crack tension force decreases in each part in turn. The rail headusually service at wear condition, so the strength of the rail head is greater than other parts and the heat treatment is also more complicated, so the crack opening force is relatively high. For the joint, however, the welding process produces a greater effect. From Figure 12, the necessary crack opening force in the rail head is smaller than the rail waist and rail bottom, and the difference is small. The necessary crack opening force is the same in the rail waist and rail bottom, which also indicates that the tensile properties of the weld area material are uniform. Comparing the same part of the base material weld, it can be seen that the crack opening force of the parent material in the rail head is twice as much as the welded rail head, which are the same in the rail waist, and the crack opening force of the bottom of the rail weld is much greater than the parent material.

The results of the da/dN-$\Delta K$ curves obtained from the traditional calculation of Paris theory and the results of the da/dN-$\Delta K$ curves obtained from the modified calculation based on Elber theory are shown in Figure 13. From the result, it can be seen that the bandwidth between the da/dN-$\Delta K$ curves obtained from the traditional calculation of Paris theory and the da/dN-$\Delta K$ curves obtained from the modified calculation based on Elber theory of the parent material specimens from the rail head to the rail waist to the rail bottom gradually becomes narrower, and there is a larger bandwidth in the test results of the material at the rail head, which indicates that there is a large plasticity-induced crack closure effect existing in the fatigue crack propagation process of the parent material rail head material, and the plasticity-induced crack closure effect of the material at the rail waist to the rail bottom gradually decreases. The bandwidth between the da/dN-$\Delta K$ curves obtained from the traditional calculation of Paris theory and the da/dN-$\Delta K$ curves obtained from the modified calculation based on Elber theory for the welded joint area material is generally smaller compared to the base material area, and the bandwidth widens slightly from the rail head to the rail waist to the bottom weld, indicating that the material in the welded joint area has less crack-tip plasticity-induced crack closure effect during fatigue crack propagation, and the deviation is smaller between da/dN-$\Delta K$ curves obtained from the traditional calculation of Paris theory and the da/dN-$\Delta K$ curves obtained from the modified calculation based on Elber theory.

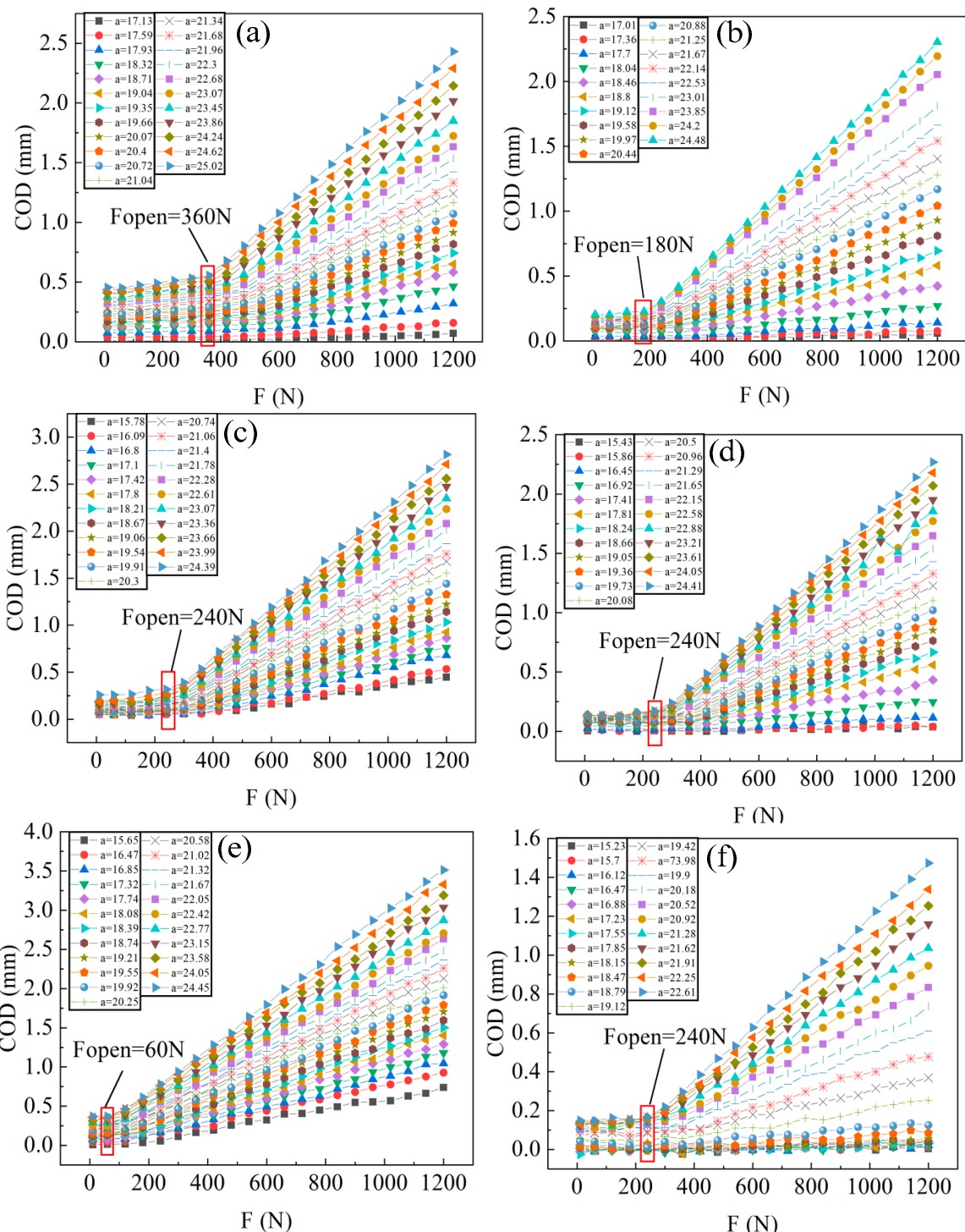

**Figure 12.** Determination of $F_{open}$: (**a**) specimen B1; (**b**) specimen W1; (**c**) specimen B2; (**d**) specimen W2; (**e**) specimen B3; (**f**) specimen W3.

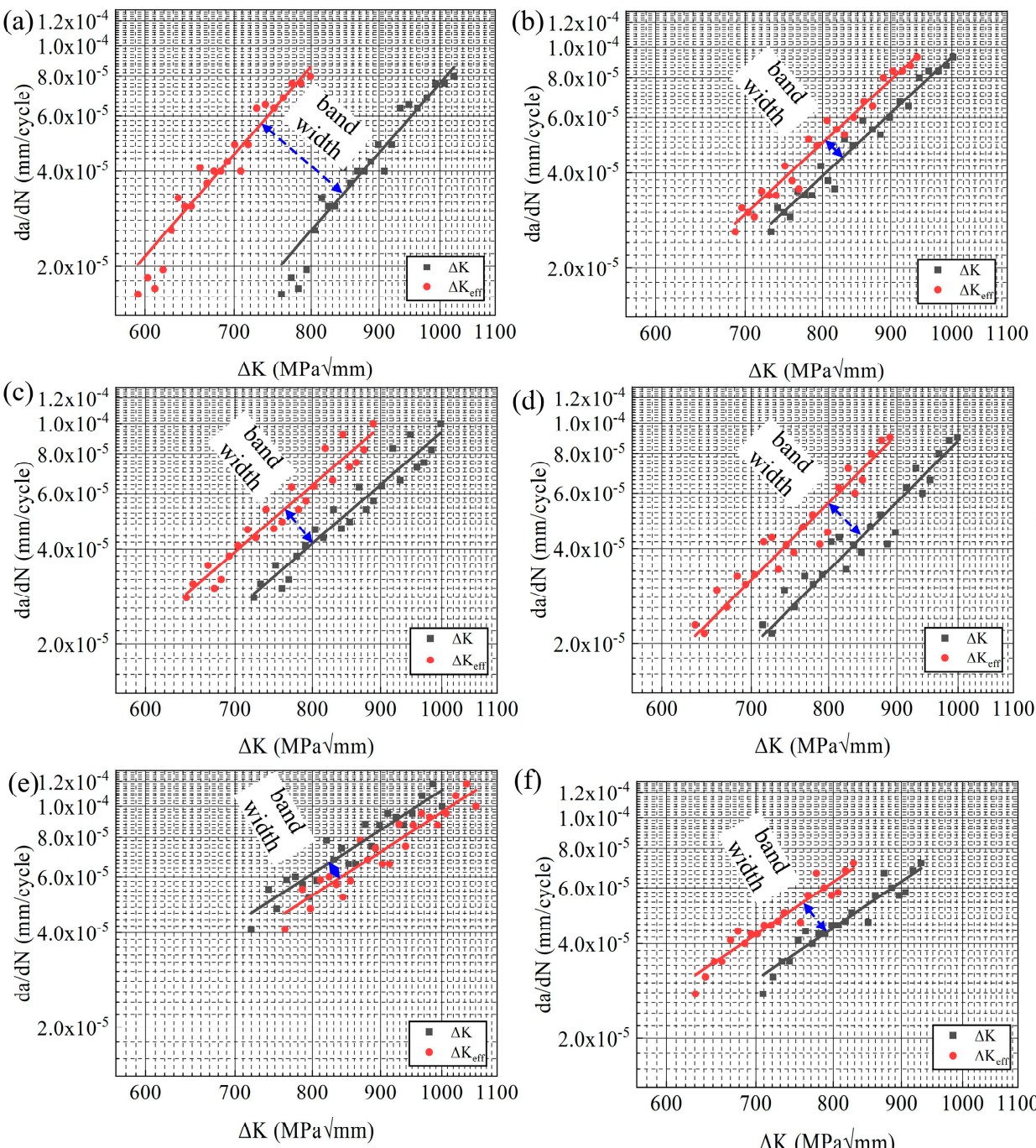

**Figure 13.** Calculation of da/dN-ΔK curves based on Paris theory and calculation of da/dN-ΔK curves based on Elber theory correction: (**a**) specimen B1; (**b**) specimen W1; (**c**) specimen B2; (**d**) W2 specimens; (**e**) B3 specimens; (**f**) W3 specimens.

Table 3 shows the values of the respective fitting parameters based on the Paris theory and the Elber theory. As can be seen from the figure, the conventional curve is exactly parallel to the Elber correction curve. However, observing the respective fitting parameters shows that m is unchanged before and after the correction, and only parameter c has changed. That is, the crack-tip plasticity-induced crack closure has a large effect on parameter *c* and no significant effect on parameter *m*.

**Table 3.** Comparison of traditional modified fitting parameters based on Paris theory and modified fitting parameters based on Elber theory.

|  | B1 | B2 | B3 | W1 | W2 | W3 |
|---|---|---|---|---|---|---|
| Value *c* by fitting Paris law | $2.808 \times 10^{-19}$ | $1.067 \times 10^{-15}$ | $6.964 \times 10^{-13}$ | $1.892 \times 10^{-16}$ | $1.925 \times 10^{-17}$ | $2.311 \times 10^{-13}$ |
| Value *m* by fitting Paris law | 4.811 | 4.272 | 2.713 | 3.897 | 4.219 | 2.855 |
| Value *c* by modified fitting | $9.285 \times 10^{-19}$ | $1.632 \times 10^{-15}$ | $8.158 \times 10^{-13}$ | $2.409 \times 10^{-16}$ | $3.147 \times 10^{-17}$ | $3.223 \times 10^{-13}$ |
| Value *m* by modified fitting | 4.811 | 4.272 | 2.713 | 3.897 | 4.219 | 2.855 |

## 4. Conclusions

This paper applied the DIC technique to characterize the fatigue crack propagation parameters for rail and welded joints, and the following main conclusions were obtained:

(1) A new method for determining crack-tip coordinates is proposed. That is, the y-coordinate of the crack tip is obtained by the relationship between the vertical displacement and the y-coordinate, the plastic zone of the crack tip is obtained by converting the DIC data, the $r_p$ value of the plastic zone dimension is calculated according to the Irwin formula, and the x-coordinate is determined by combining the already determined y-coordinate of the crack tip, the $r_p$ value, and the plastic zone profile.

(2) The precise locations of the respective CTOD measurement points for the base material and the joint were determined from the COD data, and the $\Delta$CTODs during loading and unloading of the overloaded specimens were extracted accordingly, and the $\Delta$CTOD$_{pl}$ and $\Delta$CTOD$_{el}$ were successfully calculated, which can provide some reference for the precise determination of the plastic CTOD for fatigue crack propagation of the material.

(3) The crack opening force $F_{open}$ was extracted based on the COD data obtained from the DIC test. Then, the method for calculating the amplitude of the effective stress-intensity factor for fatigue crack propagation, which considers the crack-tip plasticity-induced crack closure effect, was proposed based on the Elber theory. The crack growth-rate curves were fitted based on the conventional Paris formula and the Elber formula by considering the crack closure effect, and it was found that the crack-tip plasticity-induced crack closure effect only had an effect on parameter *c*, and had no significant effect on parameter *m*.

**Author Contributions:** X.-Y.F.: writing-original draft, validation, writing—review and editing, supervision, methodology, and funding acquisition; J.-E.G.: investigation, formal analysis, and validation; W.H.: investigation and formal analysis; J.-H.W.: writing—review and editing and validation; J.-J.D.: writing—review and editing, validation, methodology, and supervision. All authors have read and agreed to the published version of the manuscript.

**Funding:** This work was supported by the Sichuan Science and Technology Program (Grant No. 2020YFH0082), the Sichuan Science and Technology Planning Project (Grant No. 2022JDJQ0019), the China Postdoctoral Science Foundation (Grant No. 2019M653474), and the open research fund of MOE Key Laboratory of High-Speed Railway Engineering, Southwest Jiaotong University.

**Data Availability Statement:** The data presented in this study are available on request from the corresponding author.

**Conflicts of Interest:** The authors declare no conflict of interest.

## Nomenclature

*K*, the stress-intensity factor; $\Delta K$, the stress-intensity factor amplitude; $\Delta K_{eff}$, the effective stress-intensity factor amplitude; *da/dN*, fatigue crack growth rate; $F_{op}$, the crack opening force; *R*, stress ratio; $F_{max}$, the maximum load; $F_{min}$, the minimum load; *a*, crack length; *B*, specimen thickness; $r_p$, the length of the plastic zone; $\sigma_{p0.2}$, the yield strength.

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
