# Peer review of "Novel Characterizations of Effective SIFs and Fatigue Crack Propagation Rate of Welded Rail Steel Using DIC"

_metals, doi:10.3390/met13020227_

Round 1

Reviewer 1 Report

Comments:

The manuscript titled “Novel characterization of effective SIFs and fatigue crack propagation rate of welded rail steel using DIC” was investigated. The research work is of significance because it presents an approach that can be used to determine crack propagation due to fatigue, thus presenting a body of work that can advance the knowledge of crack propagation on welded rails. Minor corrections to be addressed are highlighted below:

 ·       Delete “have” after in-text references

·       Line 47: revise: “which will produce a crack closure effect”

·       Line 46-51: Change the tense

·       Line 56: Numerical number [2] is missing for reference: Elber et al. Elber is a single author, delete et al.

·       Line 62-64: revise the sentence, use locally instead of at home, include and between numerical simulation methods and experimental method.

·       Line 91-92: Consider using with to replace and “so the displacement field method is considered to be more effective and less error.

·       Line 100: Delete “therefore”

·       Line 112: Remove initial B, in the reference

·       Materials and experimental procedure

Which welding method is being considered? The HAZ is more prone to failure than the weld metal. Were the difference in properties between these regions considered?

·       Line 143: Revise

·       Line 172: Change case for the reference “Tada et al., [34]” for consistency,

·       Line 193: Delete in Figure 3; “From the results in Figure 3”

·       Line 302-306: Revise

·       Line 310: Use “at” to replace “with”

·       Line 312: Avoid using pronouns (i.e. we)

·       Line 351: Revise: “because the rail head's wear is more serious”

·       Line 354: “The results in Figure 12, not of Figure 12

·       Line 391-393: Caption and figure should fit in one page

·       Line 473: Reference 22, author name “STOYCHEV” is written out in capital letters. This is not consistent with the rest of the references in the manuscript.

·       General comment: Great work, however, the manuscript should be thoroughly edited before final submission.

Author Response

Please see the uploading file, thank you.

Reviewer 2 Report

Comments:

1. line 147 : is "During the experiment, the stress ratio R=0.1, Fmax=1200 N, and sine wave loading were used, and during the overload experiment, the stress ratio R=0.3, Fmax=2000 N, and sine wave loading were also used." When "overload experiment" starts? Is there any condition for this?

2. line 150: "14.84 mm", Why exactly is this value?

3. Chapter 2: Figure 1(c) is not referred to in the text .

4. It does not indicate when the samples presented in Figure 1(c) are used. Explain this.

5. line 177: is "W is the standard CT specimen parameter". Where is its description?

6. line 177: is "a is the crack length". From which point the crack length is calculated?

7. line 226, Figure 5, is: "Formula 3-1, 3-2, 3-3", should be equation 5, 6, 7 respectively.

8. What strength machine was used in experiment?

9. Show photo with experimental stand.

10. Figure 13. In axis Y should it not be "log da/dN"? Check it.

11. line 494: refernece nr 33 is incomplete

12. line 24: is "Fmax, the maximum load", sometimes you use "Fmax" - see eq. 2, at another time you use "Pmax" - see eq. 3. Correct it.

Author Response

Please see the uploading file, thank you!

Reviewer 3 Report

The manusrcipt contains all the required elements that characterize well-written scientific articles.

I suggest doing the nomenclatere in the form of a table and including all symbols, abbreviations and markings in it - there is quite a lot of it in the paper and it will be better to explain them at the beginning of the article.

An excellent introduction, with a properly conducted literature review.

Please do not use the words "work, works" in relation to scientific articles - preferred words are "paper, manuscript, scientific article, etc.".

The research material should rather be described with the word "specimen", not "sample".

Please work on the quality of figures 1b and 1c. It is best to prepare them in the form of vector graphics - e.g. AutoCad.

In Figure 5, please replace "Formula 3-1" with "Eq. (5)", "Formula 3-2" to "Eq. (6)", "Formula 3-3" to "Eq. (7)” – figures must be consistent with the text of the manuscript.

The authors have to correct some figures, because in the paper there are various quantities that have some subscripts - e.g. Fopen-line 362, and in Figure 12 there is the designation "Fopen". Once again, I emphasize - formulas, markings, the text of the manuscript, figures and tables must be consistent. Please correct it.

The only remark - did the authors not consider increasing numbers of the specimens, to show the reproducibility of the results - at least three for each case? Then there would be the most appropriate response of the material to the behavior under a certain load. I am asking for your attitude and possible justification for the small number of specimens in the research. In fatigue tests, especially in the field of fatigue fracture of materials, we encounter a significant dispersion of results. Please answer and post the answers in your paper.

Apart from editing and editorial corrections, I have no comments on the manuscript. I think it is valuable and worth publishing.

I ask the authors to make corrections, resubmit the paper for review. I recommend a minor revision.

Author Response

(The authors gave the same response as above.)

Round 2

Reviewer 2 Report

Comments:

1. Explain in more detail why two types of samples were used.

Type 1: Fig 1b and type 2: Fig 1c.

2. Improve quality of Fig 1 b) and c)

3. Improve equation (3), should be Fmax and Fmin 4.

What types of specimen were adopted in table 3 (from Fig 1b) or/and from Fig 1c)). Improve it. Improve column "Experimental condition". Only specimen "W4" has "R=0.3 Overload 100%" condition?

Author Response

Please see the uploaded attachment.
